# Kinesin Kip2 enhances microtubule growth *in vitro* through length-dependent feedback on polymerization and catastrophe

**Anneke Hibbel[1,6], Aliona Bogdanova[1], Mohammed Mahamdeh[2], Anita Jannasch[1,3], Marko Storch[4], Erik Schäffer[3], Dimitris Liakopoulos[5], Jonathon Howard[2]***

[1]Max Planck Institute of Molecular Cell Biology and Genetics, Dresden, Germany; [2]Department of Molecular Biophysics & Biochemistry, Yale University, New Haven, United States; [3]Zentrum für Molekularbiologie der Pflanzen, Eberhard-Karls-Universität, Tübingen, Germany; [4]Department of Life Sciences, Imperial College London, London, United Kingdom; [5]CRBM-CRNS, Montpellier, France; [6]Institute of Biochemistry, ETH Zurich, Zurich, Switzerland

**Abstract** The size and position of mitotic spindles is determined by the lengths of their constituent microtubules. Regulation of microtubule length requires feedback to set the balance between growth and shrinkage. Whereas negative feedback mechanisms for microtubule length control, based on depolymerizing kinesins and severing proteins, have been studied extensively, positive feedback mechanisms are not known. Here, we report that the budding yeast kinesin Kip2 is a microtubule polymerase and catastrophe inhibitor in vitro that uses its processive motor activity as part of a feedback loop to further promote microtubule growth. Positive feedback arises because longer microtubules bind more motors, which walk to the ends where they reinforce growth and inhibit catastrophe. We propose that positive feedback, common in biochemical pathways to switch between signaling states, can also be used in a mechanical signaling pathway to switch between structural states, in this case between short and long polymers.

**\*For correspondence:** jonathon.
howard@yale.edu

**Competing interests:** The authors declare that no competing interests exist.

## Results and discussion

The budding yeast kinesin Kip2 promotes microtubule growth in vivo. Deletion of this kinesin results in nuclear migration defects, and the phenotype is associated with shorter, less abundant cytoplasmic microtubules (*Cottingham and Hoyt, 1997*; *Huyett et al., 1998*; *Miller et al., 1998*; *Caudron et al., 2008*). Conversely, Kip2 overexpression results in hyper-elongated cytoplasmic microtubules (*Carvalho et al., 2004*). The stabilization of microtubules by Kip2 is thought to be indirect and a consequence of Kip2 transporting the growth regulator Bik1 (Clip170) to microtubule plus ends (*Carvalho et al., 2004*; *Caudron et al., 2008*).

To test whether Kip2 alone can promote microtubule growth, the activity of full-length, purified Kip2 was measured in dynamic microtubule assays using porcine tubulin in the presence of adenosine triphosphate (ATP) (*Gell, et al., 2010*; *2011*) (*Figure 1A*). Within 10 min, Kip2 (*Figure 1B, C*), as well as Kip2-enhanced green fluorescent protein (eGFP) (*Figure 1—figure supplement 1A*), strongly increased the length of freshly polymerized microtubules (p<0.0001, Welch's unpaired *t*-test, please refer to *Table 1* for porcine microtubule parameter values). The effect of Kip2 on microtubule length was almost completely inhibited when ATP was replaced by the non-hydrolyzable ATP

**eLife digest** Cells contain an extensive network of long filaments called microtubules, which are made of a protein called tubulin and are essential for a wide variety of processes such as enabling cells to divide and move. Microtubules also serve as tracks along which motor proteins transport molecules from one part of the cell to another.

In yeast cells, a motor protein called Kip2 transports its cargo to one end (known as the "plus end") of the microtubules. This is the fastest-growing end of the microtubule, although it frequently switches between phases of growth and shrinkage. Previous research has shown that cells containing reduced amounts of Kip2 have much shorter filaments than normal cells. One suggested explanation of these results is that Kip2 controls microtubule growth by transporting a protein that regulates filament length to the plus end of the microtubule.

However, by adding purified Kip2 to microtubules Hibbel et al. have now shown that Kip2 on its own can increase the length of filaments. The microtubules also switch much less frequently between growth and shrinkage in the presence of Kip2. This is because Kip2 moves along filaments at a speed that is greater than the rate at which microtubules grow. This sets up a positive feedback loop that causes microtubule growth to accelerate, as more copies of Kip2 can bind to longer microtubules. Each of these motor proteins can then move to the plus end of the microtubule and help the filament to grow even longer.

Several challenges remain. What is the molecular mechanism by which Kip2 increases the rate at which tubulin subunits are added to the microtubule end: does Kip2 carry tubulin dimers to the end in a shuttle-type mechanism? How does Kip2 prevent other proteins from promoting shrinkage? What stops microtubules from growing when they reach the end of the cell?

analog adenylyl imidodiphosphate (AMP-PNP) (*Figure 1C*, blue markers, p<0.0001), showing that growth promotion requires ATP hydrolysis. To quantify how Kip2 influences microtubule dynamics, we drew kymographs from the time-lapse images of the dynamic microtubule assay (*Figure 1D*). Kip2 increased the growth rate of microtubules (the slope of the growing microtubule in the kymograph) 2.9-fold (*Figure 1E*). In addition, Kip2 reduced the frequency of catastrophe (the transition between growth and shrinkage phases) approximately 10-fold (*Figure 1F*). Kip2 did not affect the shrinkage rate (*Figure 1—figure supplement 1B*), or the frequency of rescue (the transition between shrinkage and growth phases, *Figure 1—figure supplement 1C*). All dynamic data on porcine tubulin are contained in *Table 1*. Note that rescue is not expected to have a large effect on microtubule length in our assays. This is because at lower Kip2 concentrations (< 10 nM), the average distance shortened following catastrophe (the shrinkage rate divided by the rescue frequency) is greater than the average distance grown before catastrophe (the growth rate divided by the catastrophe frequency), so microtubules usually shrink all the way back to the seed (as expected by theory, *Verde et al., 1992*). On the other hand, at higher Kip2 concentrations, catastrophes are so rare that microtubules are expected to be very long before they catastrophe (>18 μm for [Kip2] ≥ 10 nM). Consistent with the small contribution of rescue, the measured increase in microtubule length accorded with the effects of Kip2 on the growth rate and the catastrophe frequency alone (*Figure 1C*, red line). The half-maximal stimulation of polymerization and inhibition of catastrophe occurred at ≈7 nM Kip2. Given that the cellular concentration of Kip2 is ≈25 nM (*Ghaemmaghami et al., 2003*), these results show that Kip2 affects microtubule dynamics in vitro at physiologically relevant concentrations.

To exclude potentially confounding effects introduced by using fluorescently labeled porcine brain tubulin, as well as to confirm that Kip2, which is a yeast protein, has the same activity on its conspecific protein, we repeated the dynamic microtubule assays with unlabeled yeast tubulin (*Widlund et al., 2012*) and differential interference contrast (DIC) microscopy (*Figure 1—figure supplement 2A*, please refer to *Table 2* for yeast microtubule parameter values). Consistent with our porcine tubulin results, Kip2 increased the yeast microtubule growth rate by 2.3-fold and inhibited catastrophe 20-fold (*Figure 1—figure supplement 2B,C*). The half-maximal stimulation of polymerization and inhibition of catastrophe for yeast tubulin occurred at ≈12 nM Kip2, similar to the

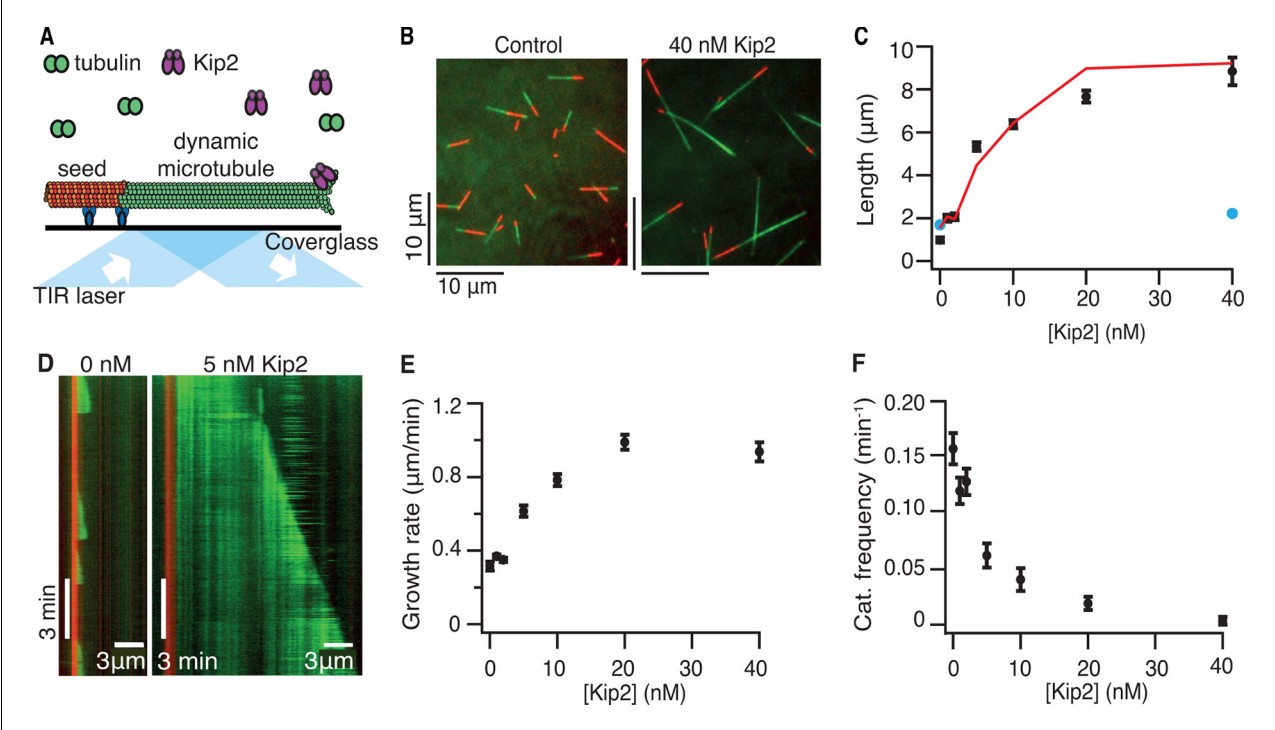

**Figure 1.** Kip2 is a microtubule polymerase and an anti-catastrophe factor for porcine tubulin. (**A**) Schematic of the experimental design: porcine tubulin (green) polymerizes onto stabilized microtubules (red) bound to the coverslip with antibodies (blue), imaged using TIRF microscopy. (**B**) Microscopy images of dynamic microtubules grown from stabilized seeds without (left) and with 40 nM Kip2 (right) at $t$ = 10 min. (**C**) Microtubule length as a function of Kip2 concentration in ATP (black circles) or AMP-PNP (blue circles) at $t$ = 10 min. The red line indicates the expected microtubule length at $t$ = 10 min, calculated from the measured growth rates and catastrophe frequencies in **Table 1** according to the formula $L = (v_+/f_{+-})[1-\exp(-tf_{+-})]$, where $v_+$ is the growth rate and $f_{+-}$ is the catastrophe frequency (ignoring rescues and assuming that regrowth occurs without delay). (**D**) Kymographs showing typical microtubule growth without (left) and with 5 nM Kip2 (right) in ATP. (**E**) Microtubule growth rate as a function of Kip2 concentration in ATP. (**F**) Catastrophe frequency as a function of Kip2 concentration in ATP. All error bars are standard errors of the mean. Please refer to **Table 1** for values. AMP-PNP, adenylyl imidodiphosphate; ATP, adenosine triphosphate; TIRF, total internal reflection fluorescence.

The following figure supplements are available for figure 1:

**Figure supplement 1.** Kip2 has no significant effect on microtubule shrinkage rate or rescue frequency.

**Figure supplement 2.** Kip2 is a microtubule polymerase and an anti-catastrophe factor for yeast tubulin.

**Figure supplement 3.** Kip2 increases the growth rate in GTP-tubulin and lowers the off rate of GMPCPP tubulin.

**Figure supplement 4.** SDS–PAGE gels of Kip2 and Kip2-eGFP.

concentration at which Kip2 regulates porcine brain microtubules. In summary, Kip2 is a microtubule polymerase and anti-catastrophe factor in vitro and does not require additional proteins such as Bik1 for these activities. We will defer discussing a potential role for Bik1 until the end of the manuscript.

To gain insight into the mechanism of Kip2's polymerase and anti-catastrophe activities, we determined how Kip2 affects microtubule assembly and disassembly kinetics. By measuring the rate of growth of porcine microtubules from guanylyl (α,β)methylene-diphosphonate (GMPCPP) seeds over a range of tubulin concentrations (**Figure 1—figure supplement 3A**), we found that Kip2 doubled the effective tubulin association rate constant ($k_{on}$, the rate that tubulin is stably incorporated into the microtubule lattice) to 1.5 $\mu M^{-1} \cdot s^{-1}$ (at the plus end) from 0.7 $\mu M^{-1} \cdot s^{-1}$ in the absence of Kip2. In addition to accelerating the net addition of subunits, Kip2 also facilitated microtubule nucleation on the seeds, with robust growth observed at tubulin concentrations as low as 4 $\mu M$, compared with 10 $\mu M$ in the absence of Kip2 (**Figure 1—figure supplement 3A**). Thus, Kip2 acted like a nucleation

**Table 1.** Parameters of microtubule dynamics for 12 μM porcine tubulin (mean ± SE).

| [Kip2] (nM) | Length (μm) | Growth rate (μm/min) | Catastrophe frequency (min[-1]) | Catastrophe distance[a] (μm) | Shrinkage rate (μm/min) | Rescue frequency (min[-1]) | Rescue distance[b] (μm) |
|---|---|---|---|---|---|---|---|
| 0 | 1.0 ± 0.1 (n = 54) | 0.32 ± 0.02 (n = 172) | 0.166 ± 0.015 (n = 126) | 1.9 ± 0.2 | 27.6 ± 1.0 (n = 130) | 0.88 ± 0.33 (n = 7) | 32 ± 12 |
| 1 | 1.0 ± 0.2 (n = 75) | 0.37 ± 0.01 (n = 152) | 0.126 ± 0.012 (n = 104) | 2.9 ± 0.3 | 27.7 ± 0.9 (n = 85) | 0.20 ± 0.14 (n = 2) | 140 ± 100 |
| 2 | 2.1 ± 0.2 (n = 88) | 0.35 ± 0.01 (n = 159) | 0.135 ± 0.013 (n = 115) | 2.6 ± 0.3 | 29.7 ± 0.9 (n = 110) | 0.54 ± 0.24 (n = 5) | 55 ± 25 |
| 5 | 5.3 ± 0.2 (n = 82) | 0.62 ± 0.03 (n = 77) | 0.065 ± 0.011 (n = 33) | 10 ± 2 | 28.4 ± 1.8 (n= 45) | 2.2 ± 0.7 (n = 9) | 13 ± 4 |
| 10 | 6.4 ± 0.2 (n = 75) | 0.78 ± 0.03 (n = 38) | 0.043 ± 0.011 (n = 16) | 18 ± 5 | 29 ± 4 (n = 18) | 1.8 ± 0.7 (n = 6) | 16 ± 6 |
| 20 | 7.7 ± 0.3 (n = 68) | 0.99 ± 0.04 (n = 36) | 0.020 ± 0.006 (n = 10) | 50 ± 16 | - | -. | - |
| 40 | 8.8 ± 0.6 (n = 26) | 0.94 ± 0.05 (n = 18) | 0.004 (n = 1) | 235 | - | - | - |

[a]The catastrophe distance is the growth rate divided by the catastrophe frequency.
[b]The rescue distance is the shrinkage rate divided by the rescue frequency.

factor in analogy to XMAP215 (*Wieczorek et al., 2015*). The increased growth rate in the presence of Kip2 is expected to have only a modest effect on catastrophe because doubling the rate of microtubule growth by doubling the tubulin concentration only decreases the catastrophe frequency about twofold (*Gardner et al., 2011*; *Walker, et al., 1988*). Our observation that the catastrophe frequency decreased 10-fold might be explained by our finding that 40 nM Kip2 decreased the rate of dissociation of GMPCPP-tubulin subunits from GMPCPP microtubules ($k_{off}$) approximately three-fold (*Figure 1—figure supplement 3B*). If GMPCPP-tubulin acts as an analog for guanosine-5'-triphosphate (GTP)-tubulin (*Hyman et al., 1992*), then a decrease in $k_{off}$ is expected to stabilize the GTP cap and therefore inhibit catastrophe (*Bowne-Anderson et al., 2013*; *Coombes et al., 2103*; *Margolin et al., 2011*). Thus, the increase in $k_{on}$ and the decrease in $k_{off}$ likely account for most of the decrease in the catastrophe frequency.

To determine how Kip2 targets the plus ends of microtubules, we characterized its biophysical properties in single-molecule motility assays (*Figure 2A*). Kymographs revealed that in 1 mM ATP, single Kip2-eGFP molecules associated with GMPCPP-stabilized porcine microtubules along the lattice and walked processively toward the plus end of the microtubule (*Figure 2B*, *Figure 2—figure supplement 1A*). The velocity was 5.0 ± 0.9 μm/min at 28°C (mean ± standard deviation [SD], n = 674 traces). The average run distance before dissociating was 4.1 ± 0.3 μm (mean ± SE, n = 217, *Figure 2—figure supplement 1B*). A similar velocity was observed by *Roberts et al. (2014)*, though the run distance was shorter (1.2 μm). At the plus end, Kip2-GFP resided for 30 ± 26 s before dissociating (mean ± SD, n = 40, *Figure 2D*, *Figure 2—figure supplement 1C*), leading to an accumulation of up to 12 Kip2-eGFP molecules at the plus-end, based on the fluorescence intensity (*Figure 2B*). When the ATP was replaced by the non-hydrolyzable analog AMP-PNP, Kip2-eGFP tightly bound to the lattice and did not translocate (*Figure 2C*). Kip2-eGFP moved slower on dynamic microtubules (2.1 ± 0.89 μm/min), but this velocity is still greater than the microtubule's growth speed, so Kip2-eGFP was able to catch up to the growing ends of dynamic microtubules and track them (*Figure 2E*). Based on these properties, we conclude that Kip2's mechanism differs from that of the well-studied microtubule polymerase XMAP215. XMAP215 targets ends by diffusion and capture (*Brouhard et al. 2008*; *Widlund et al. 2011*), increases both $k_{on}$ and $k_{off}$ (*Brouhard et al., 2008*) and has little effect on catastrophe (*Zanic, et al., 2013*). Furthermore, Kip2's ATPase activity is necessary for its activity, while XMAP215 is not an ATPase. Thus, Kip2 is a unique regulator of microtubule dynamics. Two models for growth promotion can be envisaged. Kip2 may increase growth rates by shuttling tubulin to the microtubule plus end, locally increasing the tubulin concentration. Alternatively, it could promote microtubule growth by acting as a processive polymerase

**Table 2.** Parameters of microtubule dynamics for 4 $\mu$M yeast tubulin (mean ± SE).

| [Kip2] (nM) | Growth rate (µm/min) | Catastrophe frequency (min⁻¹) |
|---|---|---|
| 0 | 0.257 ± 0.004 (n = 300) | 0.234 ± 0.017 (n = 191) |
| 5 | 0.302 ± 0.005 (n = 263) | 0.137 ± 0.012 (n = 141) |
| 10 | 0.353 ± 0.008 (n = 146) | 0.103 ± 0.010 (n = 116) |
| 20 | 0.572 ± 0.01 (n = 57) | 0.020 ± 0.006 (n = 13) |
| 40 | 0.589 ± 0.012 (n = 48) | 0.009 ± 0.004 (n = 5) |

while at the plus end, similar to XMAP215 (*Brouhard et al., 2008*). More work will be required to distinguish between these and other mechanisms.

To probe the mechanical properties of Kip2, we measured the stall force of single molecules using optical tweezers (*Jannasch et al., 2013*). Positional tracking of single Kip2-powered microspheres moving along GMPCPP-stabilized porcine microtubules as a function of time under constant load revealed a zero-force speed of 4.0 ± 0.5 µm/min at 24.5°C, similar to that measured in the total internal reflection fluorescence (TIRF) assays. Kip2 stalled at a force of 0.81 ± 0.04 pN (*Figure 3A*) and showed a nearly linear force-velocity relation with increased velocity as the assisting force was increased (*Figure 3C*). At high forces, the motor often slipped along the microtubule in the direction of the applied force without detaching (*Figure 3B*). The ability to switch from the slip state to the normal translocation mode is thought to increase processivity by linking together several shorter run lengths (*Jannasch et al., 2013*). Thus, Kip2 is a processive, low-force motor with long run lengths

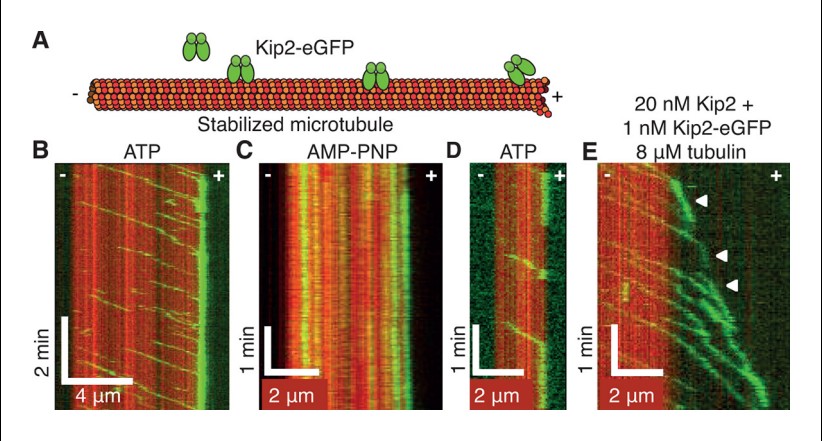

**Figure 2.** Kip2 is a highly processive motor that dwells at plus ends. (A) Schematic of the experimental design. (B) Kymograph showing processive motility and plus end accumulation of individual Kip2-eGFP molecules on GMPCPP-stabilized microtubules in 1 mM ATP. The concentration of Kip2-eGFP was 0.085 nM. (C) Kymograph showing tightly bound Kip2-eGFP molecules in AMP-PNP. (D) Kymograph showing end residence of individual Kip2-eGFP molecules in ATP. (E) Kymograph showing end residence of 1 nM Kip2-eGFP spiked into 20 nM unlabeled Kip2 in the presence of 8 µM unlabeled tubulin in ATP. Arrow heads indicate microtubule plus end tracking events. ATP, adenosine triphosphate; AMP-PNP, adenylyl imidodiphosphate; eGFP, enhanced green fluorescent protein; GMPCPP, guanylyl (a,ß)methylene-diphosphonate.

The following figure supplement is available for figure 2:

**Figure supplement 1.** Kip2-eGFP velocity, run length and end residence time distributions.

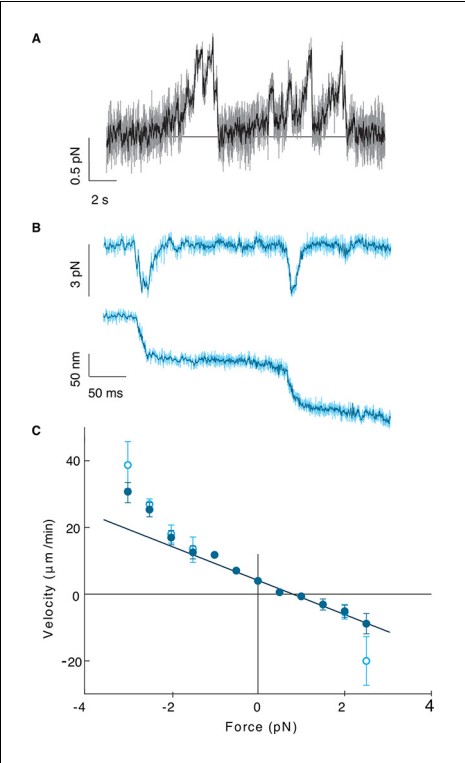

**Figure 3.** Kip2 is a low-force motor. (**A**) Stall force measurement tace. Sampling rate: 10 kHz, raw data (gray), boxcar filtered to 50 Hz (black). A force of 0.5 N corresponds to a displacement of about 17 nm. (**B**) Time trace for a slip event under 3 pN assisting force. Sampling rate: 20 kHz, raw data (light cyan), boxcar filtered to 400 Hz (dark cyan). (**C**) Kip2 force-velocity curve: positive is a hindering (load) force and negative is an assisting force. Open symbols include slip events. Error bars are standard errors of the mean.

and end residence times. The low force supports the idea that individual Kip2 motors transport small cargos such as dynein, Bik1, and other molecules (*Roberts et al., 2014*) rather than organelles, although it is possible that multiple Kip2s could cooperate to move larger cargos. The strong localization to the microtubule plus end accords with Kip2 being a regulator of microtubule dynamics.

The low force and high processivity of Kip2 are reminiscent of the microtubule depolymerase Kip3, in the kinesin-8 family (*Jannasch et al., 2013*; *Varga et al., 2006*). Kip3 is a length-dependent depolymerase that uses an antenna mechanism to preferentially localize to the plus ends of longer microtubules (*Varga et al., 2009*). We therefore tested whether the promotion of microtubule growth by Kip2 is length-dependent (*Figure 4A,B*). Without Kip2, microtubules grew at a length-independent, constant rate (black circles, p=0.06, Student's *t*-test on a linear fit to the raw data). By contrast, at low (1–2 nM) and intermediate (5–10 nM) Kip2 concentrations, long microtubules grew faster than short microtubules (p<0.0001). At high Kip2 concentrations (20–40 nM), microtubules again grew at a constant, length-independent rate (green circles, p=0.36); however, at these high Kip2 concentrations, we expect all the length dependence to be in the first few microns, which is not well resolved in these experiments (see green fitted line). An analysis of yeast microtubule growth rates as a function of microtubule length yielded similar results (*Figure 4—figure supplement 1A*). Thus, Kip2 is a length-dependent microtubule polymerase.

To test whether Kip2 also prevents catastrophe in a length-dependent manner, we measured porcine microtubule lengths at the moment of catastrophe. To compare the catastrophe frequency at short versus long microtubule lengths, we set a cut-off length at 4 μm, which equals the run length of Kip2. Using data from the dynamic microtubule assays (*Figure 1*), we measured the catastrophe length for short microtubules as the total distance that microtubules grew while their length (including seed) was shorter than 4 μm divided by the number of catastrophes that occurred at lengths < 4 μm. For long microtubules, we summed the distance that microtubules grew while longer than 4 μm

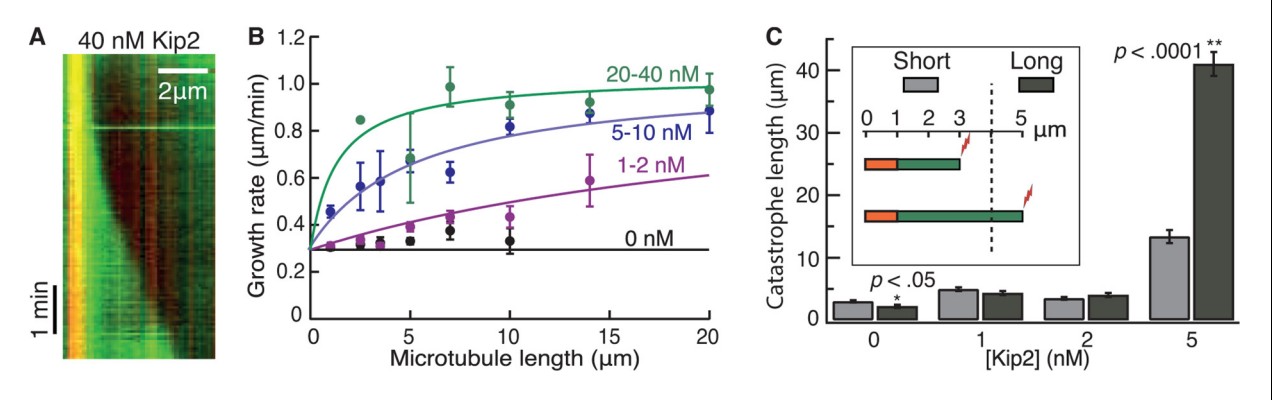

**Figure 4.** Kip2 promotes porcine microtubule growth in a length-dependent manner. (**A**) Kymograph showing acceleration of microtubule growth with increasing length at 40 nM Kip2. (**B**) Porcine microtubule growth rate as a function of length without Kip2 (black) and binned for 1–2 nM Kip2 (purple), 5–10 nM Kip2 (blue) and 20–40 nM Kip2 (green). Lengths are binned for 0–2 μm, 2–3 μm, 3–4 μm, 4–6 μm, 6–8 μm, 8–12 μm, 12–16 μm and 16–24 μm. The data were fit with the equation, where $v_0 = 0.294 \pm 0.009$ μm/min is the initial growth rate; $v_{max} = 1.03 \pm 0.03$ μm/min is the maximum growth rate; $L$ is microtubule length and $A = 39.8 \pm 5.4$ μm·nM. (**C**) Mean catastrophe length at various Kip2 concentrations for short (light gray) and long (dark gray) microtubules. In the short microtubule bins, we summed the total distance that microtubules grew while shorter than 4 μm and divided by the number of catastrophes that occurred at lengths < 4 μm. In the long microtubule bin, we summed the total distance that microtubules grew while longer than 4 μm and divided by the number of catastrophes that occurred at lengths > 4 μm (**Figure 4C**, inset). The number of catastrophes was 120 (0 nM Kip2), 102 (1 nM Kip2), 111 (2 nM Kip2) and 23 (5 nM Kip2). The number of catastrophes at higher Kip2 concentrations was too small to make statistically significant comparisons. Error bars are standard errors of the mean.

The following figure supplement is available for figure 4:

**Figure supplement 1.** Length dependence of growth and catastrophe for yeast tubulin.

(final length minus 4 μm) and divided by the number of catastrophes that occurred at lengths > 4 μm (**Figure 4C**, inset). In the absence of Kip2, the catastrophe length of longer microtubules was less than that of shorter microtubules (**Figure 4C,** 0 nM Kip2, p<0.05, Welch's unpaired *t*-test). This reflects an increase in catastrophe frequency with length, as expected due to microtubule aging (**Gardner et al., 2011**). By contrast, at 5 nM Kip2, the catastrophe length of longer microtubules was greater than that of shorter microtubules (**Figure 4C,** p<0.0001). This indicates that the inhibition of catastrophe is length-dependent. Similar results were obtained for yeast microtubules at 10 nM Kip2 (**Figure 4—figure supplement 1B,** p<0.0001). Hence, in the absence of Kip2, the catastrophe frequency increased with increasing microtubule length, whereas in the presence of Kip2, the catastrophe frequency decreased with increasing microtubule length. Thus, both the increase in microtubule growth rate and the prevention of catastrophe by Kip2 increase with increasing microtubule length.

Summarizing our results, we have found that budding yeast kinesin Kip2 promotes microtubule growth in vitro in a length-dependent manner. Because the rate at which Kip2 translocates exceeds the speed of microtubule growth, Kip2 catches up with the growing end of the microtubule (**Figure 2E**) where it promotes growth and inhibits catastrophe. As a consequence, this length dependence leads to positive feedback: the longer the microtubule, the greater the number of motors that land on it (the microtubule acts as an antenna), the higher the number of motors that can reach the plus end, and the higher the growth rate and lower the catastrophe frequency. This, in turn, leads to longer microtubules, which attract more Kip2 and so on. Hence, we expect that once a microtubule is long enough, it will effectively "escape" catastrophe and keep growing almost indefinitely, switching from a catastrophe length of only a few microns in the absence of Kip2 to a length $\geq$ 40 μm at high Kip2 concentrations (**Figure 4C**, **Table 1**). In this sense, Kip2 "paves its own way". Thus, by combining processivity with polymerase activity, Kip2 can perform an elementary 'computation' that switches short microtubules to long ones. This computation differs from that performed by kinesin-8, a length-dependent depolymerase, which stabilizes microtubule length through

negative feedback (*Gupta et al., 2006*; *Mayr et al., 2007*; *Stumpff et al., 2008*; *Su, et al., 2013*; *Widlund, et al., 2011 Varga et al., 2006*; *2009*).

We propose that this positive feedback mechanism may operate in vivo and account for the phenotype of Kip2 deletion, which is a reduction in the length and the number of cytoplasmic microtubules. First, in vivo the rate at which Kip2 translocates exceeds the rate of microtubule growth; the respective rates are 6.6 (*Carvalho et al., 2004*) and 2.3 µm/min (*Caudron et al., 2008*). Second, the run length of Kip2 (≈4 µm) exceeds the length of cytoplasmic microtubules (≈2 µm [*Caudron et al., 2008*]). Taken together, these two observations imply that almost every Kip2 that lands on a microtubule will reach the growing plus end. By promoting growth and inhibiting catastrophe, Kip2 can deliver cytoplasmic dynein (*Roberts et al., 2014*) to the distal cortex of the growing daughter bud before the microtubules catastrophe.

While the polymerase and anti-catastrophe activities can account for the deletion phenotype of Kip2, it is not obvious why microtubule hyperelongation when Kip2 is overexpressed should require Bik1 (*Carvalho et al., 2004*). We propose that Bik1 may be required to increase Kip2's processivity in vivo. Feedback can only operate if the run length exceeds the microtubule length. In our in vitro assays, the Kip2 run length was ≈4 µm, whereas that measured by *Roberts et al. (2014)* was only ≈1 µm. We do not know why there was a difference, as the assay buffers were similar. Importantly, though, *Roberts et al. (2014)* found that Bik1 could increase Kip2's run length 3–4 fold (in the presence of Bim1). Therefore, if the run length of Kip2 in vivo is short, then the requirement for Bik1 in the overexpression assays may be due to Bik1 acting as a processivity factor that increases the run length, thereby allowing more Kip2 to reach the end where it enhances microtubule growth.

## Materials and methods

### Protein purification and preparation

Porcine brain tubulin was purified and labeled with tetramethylrhodamine or Alexa Fluor 488 (Invitrogen, Carlsbad, CA) according to the standard protocols, as previously described (*Gell et al., 2011*). Preparation of GMPCPP-stabilized microtubule seeds was performed as previously described (Gell et al., 2010). Full length 6xHis-Kip2 and 6xHis-Kip2-eGFP were expressed in SF+ cells using baculovirus expression and purified using affinity chromatography over 1 ml His-affinity columns (GE Healthcare, Chalfont St. Giles, UK). Cells were lysed in 50 mM $NaH_2PO_4$, 300 mM NaCl, 0.1% Tween-20, 10 mM imidazole, protease inhibitors, 2 mM Mg-ATP, at pH = 8.0. The wash buffer consisted of 50 mM $NaH_2PO_4$, 300 mM NaCl, 100 mM imidazole, 2 mM Mg-ATP, at pH = 8.0. The elution buffer consisted of 50 mM $NaH_2PO_4$, 300 mM NaCl, 300 mM imidazole, 2 mM Mg-ATP, at pH = 8.0. Affinity column purification success was checked by sodium dodecyl sulfate polyacrylamide gel electrophoresis (SDS–PAGE) and Western blot using anti-6xHis antibody (Genscript, Piscataway, NJ). Next, the 6xHis-tags were cleaved from the protein using PreScission protease (GE Healthcare). The protease was added to the 300 mM imidazole elution fraction in a 1:50 dilution and incubated overnight on a rotary wheel at 4°C. Protein stability was confirmed by SDS–PAGE and enzymatic cleavage of the 6xHis-tag from the protein of interest by Western blot using anti-6xHis-antibody. Finally, Kip2 and Kip2-eGFP were purified to homogeneity by gel filtration over a Sephadex 200 column that was pre-washed with protein storage buffer: 1x BRB80 (80 mM PIPES, 1 mM $MgCl_2$, 1 mM EGTA, pH 6.8) supplemented with 10% glycerol, 1 mM Mg-ATP, 1 mM dithiothreitol (*Figure 1— figure supplement 4*). Final protein purity was checked by mass spectroscopy at the MPI-CBG in house mass spectroscopy facility. Protein concentration was determined by Bradford assay and purified proteins were snap-frozen using liquid nitrogen and stored at –80°C.

### Microscopy assays and imaging conditions

The dynamic microtubule assay for dynamic growth of Alexa Fluor 488-labeled tubulin from tetramethylrhodamine-labeled GMPCPP-stabilized porcine tubulin seeds were imaged by TIRF microscopy as described previously (Gell et al., 2010). The imaging buffer contained 1x BRB20 (20 mM PIPES, 1 mM $MgCl_2$, 1 mM EGTA, pH 6.8) supplemented with 100 mM KCl, 20 mM glucose, 20 µg/ml glucose oxidase, 8 µg/ml catalase, 0.1 mg/ml casein, 1 mM dithiothreitol, 0.001% tween-20, 1 mM GTP and 1 mM Mg-ATP or AMP-PNP. The single-molecule motility assay on tetramethylrhodamine-labeled GMPCPP-stabilized tubulin seeds imaged by TIRF microscopy was described previously (Gell

et al., 2010). For all experiments, the imaging buffer contained no added GTP. Imaging was performed with an Andor iXon camera on a Zeiss (Oberkochen, Germany) Axiovert 200M microscope with a Zeiss ×100/1.46 plan apochromat oil objective and standard filter sets. An objective heater (Zeiss) was used to warm the sample to 28°C.

The rate of photobleaching in our TIRF assays was low. In the AMP-PNP experiments (e.g. *Figure 2C*), the mean time to bleaching of Kip2-eGFP was 249 ± 68 s (mean ± SD, *n* = 10). Given that the average run length of 4.1 µm corresponds to a run time of 82 s (at 50 nm/s), we expect bleaching to have only a small effect on the measured run times. Similarly, bleaching will have little effect on the end residence times. The low rate of photobleaching accords with our earlier quantification of photobleaching (*Varga et al., 2009*).

DIC microscopy was described previously (*Bormuth et al., 2007*). All experiments were performed at least three times on three different days. Image analysis was performed by creating kymographs of microtubule growth events in image J. For growth and shrinkage rates, typically > 20 microtubules were measured, and the mean and standard error of the mean (SE) are reported in the text and figures. For the catastrophe frequency, we divided the total number of events by the total observation time. For the rescue distance, we divided the total observed distance that microtubules shrank by the total number of rescue events. The relative error (SE) was estimated as the inverse of the square root of the number of events. This assumes that the catastrophe and rescue events are single-step (Poisson) processes. However, if the events are multistep (e.g. from a gamma distribution), as is known to be the case for catastrophe (*Gardner et al., 2011*), then the actual SE is smaller than the calculated one.

## Optical tweezers assay preparation

Flow-cell construction and immobilization of GMPCPP-stabilized porcine microtubules were performed as previously described (*Jannasch et al., 2013*). The imaging buffer for optical tweezer experiments contained 1xBRB20 supplemented with 100 mM KCl, 20 mM glucose, 20 µg/ml glucose oxidase, 8 µg/ml catalase, 0.1 mg/mlcasein, 0.5% b-mercaptoethanol, 1 mM Mg-ATP. The channels were rinsed with 20 µl imaging buffer with Kip2-functionalized microspheres. For the Kip2-functionalized microspheres, carboxylated polystyrene microspheres (mean diameter 0.59 µm, Bangs Lab, Fishers, IN) were bound covalently to anti-GFP antibody via a 3 kDa polyethylene glycol (PEG) linker, which, in turn, bound to the C-terminal eGFP of Kip2-eGFP-6xHis, as previously described (*Jannasch et al., 2013*). The measurements were performed at 24.5°C and under single-molecule concentrations where only one out of four microspheres showed motility.

## Optical tweezers trapping experiments

Measurements were performed in a single-beam optical tweezers setup as previously described (*Schäffer et al., 2007*; *Bormuth et al., 2009*; *Jannasch et al., 2013*). All measurements were done with a trap stiffness of 0.03 pN/nm. The optical trap was calibrated by analysis of the height-dependent power spectrum density as described previously (*Tolić-Nørrelykke et al., 2006*). The force-velocity curve was measured using the constant-force mode. In this mode, the trapping laser was moved with a piezo mirror relative to the sample with an update rate of 200 Hz. Overall, we measured and analyzed the motion of 11 different single Kip2-eGFP-6xHis molecules. Data analysis was previously described (*Jannasch et al., 2013*).

## Acknowledgements

We thank T Hyman, S Diez and S Alberti for guidance; M Podolski for the kind gift of yeast tubulin; J Alper for critical reading of earlier versions of the manuscript; H Petzold for technical help with protein expression and assays; and members of the Howard and Diez laboratories for discussions, reading, and feedback. We would like to thank the following services and facilities of the MPI-CBG for their support: protein expression, chromatography and mass spectrometry. There are no potential conflicts of interest. Research reported in this publication was supported by the Max Planck Society, an European Research Council Starting Grant 2010 (Nanomech 260875) to AJ and ES, and by the National Institute of General Medicine Sciences of the National Institutes of Health under award number R01GM110386 to JH. The content is solely the responsibility of the authors and does not necessarily represent the official views of the National Institutes of Health.

## Additional information

### Funding

| Funder | Grant reference number | Author |
|---|---|---|
| European Research Council | NanoMech260875 | Erik Schäffer<br>Anita Jannasch |
| Max-Planck-Gesellschaft | | Anita Jannasch<br>Erik Schäffer |
| National Institutes of Health | R01GM110386 | Jonathon Howard |

The funders had no role in study design, data collection and interpretation, or the decision to submit the work for publication.

### Author contributions

AH, Conception and design, Acquisition of data, Analysis and interpretation of data, Drafting or revising the article, Contributed unpublished essential data or reagents; AB, Acquisition of data, Drafting or revising the article, Contributed unpublished essential data or reagents; MM, Acquisition of data, Drafting or revising the article; AJ, Conception and design, Acquisition of data, Analysis and interpretation of data; MS, Conception and design, Acquisition of data, Drafting or revising the article; ES, DL, Conception and design, Analysis and interpretation of data, Drafting or revising the article; JH, Conception and design, Acquisition of data, Analysis and interpretation of data, Drafting or revising the article

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
