## [Decision Letter]

Thank you for submitting your work entitled "Processive motility by the polymerizing kinesin Kip2 enhances microtubule growth through positive feedback" for peer review at *eLife*. Your submission has been favorably evaluated by Randy Schekman (Senior editor), a Reviewing editor, and two reviewers.

The reviewers have discussed the reviews with one another and the Reviewing editor has drafted this decision to help you prepare a revised submission.

Summary:

There is a general agreement among the reviewers that this is an informative manuscript that proposes an interesting working model for Kip2. The manuscript is therefore potentially suitable for *eLife*. However, the reviewers identified two crucial points that must be addressed in a revised manuscript.

Essential revisions:

1) The physiological significance of the activity described here for Kip2 is currently unclear, because the previous work of Carvalho et al. (2004) that over-expression of Kip2 promotes microtubule overgrowth in yeast cell, cited by the authors, was shown to be entirely dependent on the cargo of Kip2-the microtubule plus end-tracking protein Bik1 (see Figure 6 of Carvalho et al.). In a *bik* deletion mutant (but not in a deletion of the major plus-end tracker Bim1/EB1), the overexpression of Kip2 failed to promote microtubule overgrowth. This point needs to be mentioned in the Conclusion section of the manuscript. In addition, the title and Abstract overstate the data given this prior result – there is no current evidence for positive feedback operating in a cellular context and, while this idea could be included as a final Discussion paragraph, the title and Abstract should summarize the actual data in the manuscript – which is in vitro length-dependent promotion of growth and suppression of catastrophe by a processive kinesin. Of course, if this point could be addressed experimentally by including Bik1 in the assay, it would greatly improve the significance of the study. However, we realise that this may be difficult, and the reviewers would be happy with text edits that appropriately address this point.

2) A quite crucial point, that could be addressed experimentally, is whether there is any difference in force-dependence of Kip2/MT interactions at the tips of MTs relative to the lattice. If accumulation at the tip is an important property of Kip2's function, perhaps its ability to withstand loads at the tip is higher than that along the lattice. If the authors elected to carry out this experiment, their model would be supported more strongly.

Minor points:

1) A gel showing the purified Kip2 protein used in the experiments should be included in the primary figure or supplement. I could not find details on the yeast tubulin used – how was this prepared? N.B. the yeast tubulin experiments are a terrific addition to the manuscript.

2) While it is clear from the kymograph in Figure 2 that Kip2 is a processive motor protein, the authors should present the distribution of run lengths and velocities in a supplementary figure.

3) Figure 1: Previous normal reported growth rate of MTs at 12 μm (at 25 degrees) is 15 nm/s, or 0.9 um/min (Gardner, Zanic, Cell 2011). Why is the growth rate at 0 nM Kip2 half the prior value? Is it a temperature difference between the different experiments or difference in the tubulin preparations?

4) Please include the residence time distribution of single Kip2 motors as a supplementary figure. How are the single-molecule plus ends residence times determined if there are multiple Kip2s present? Are shorter MTs used to establish single-molecule conditions at the plus end for step bleaching? How is loss due to dissociation vs. loss due to bleaching discriminated? More technical detail on this should be included in the Methods.

5) Results and Discussion: "The low force argues against a role of Kip2 as an organelle transporter". The authors should tone down this statement. Organelles are often transported by multiple motors, the force measurements are performed on single Kip2 motors (stall force 0.8 pN, reported slip event 3 pN), which could very well be sufficient for organelle transport, especially if teams of motors are operating and if assumed that forces produced by groups of motors are (at least partially) additive. Also, as noted in the major comment above, the effect of Kip2 on microtubule growth appears to be dependent on its cargo Bik1 (not an organelle but still related to transport).

6) Figure 4 legend: The authors state: “Thus there was 5 μm of growth at short lengths”. Short lengths are defined < 4 μm. Please revise the text to increase clarity. This section is confusing to read when combined with the figure schematic. Both the schematic and text need revision.

7) The mechanical properties of Kip2 are likely defined using porcine tubulin. Please mention this in the text.

8) Figure 1—figure supplement 1: Res. frequency is reported in μm-1. Please correct.

9) Table 2: Please add units.

10) The prior work from the Howard group (Varga et al.) on length-dependent control focused on the processive motor Kip3 that induces subunits loss at ends. In a dynamic microtubule assay with kinesin-1 is there any effect on growth rate? Does Kip2 have higher affinity for tubulin dimers – one model would be that the unbound head at the end would help "deliver" a tubulin dimer and thereby accelerate growth (this would not explain the effect on catastrophe, however). Some discussion on what may make Kip2 special based on its catalytic domain/other features would be useful to include.

---

## [Author Response]

*Essential revisions: 1) The physiological significance of the activity described here for Kip2 is currently unclear, because the previous work of Carvalho et al. (2004) that over-expression of Kip2 promotes microtubule overgrowth in yeast cell, cited by the authors, was shown to be entirely dependent on the cargo of Kip2-the microtubule plus end-tracking protein Bik1 (see Figure 6 of Carvalho et al.). In a* bik *deletion mutant (but not in a deletion of the major plus-end tracker Bim1/EB1), the overexpression of Kip2 failed to promote microtubule overgrowth. This point needs to be mentioned in the Conclusion section of the manuscript. In addition, the title and Abstract overstate the data given this prior result – there is no current evidence for positive feedback operating in a cellular context and, while this idea could be included as a final Discussion paragraph, the title and Abstract should summarize the actual data in the manuscript – which is in vitro length-dependent promotion of growth and suppression of catastrophe by a processive kinesin. Of course, if this point could be addressed experimentally by including Bik1 in the assay, it would greatly improve the significance of the study. However, we realise that this may be difficult, and the reviewers would be happy with text edits that appropriately address this point.*

We have included “in vitro” in the title and Abstract and reworded the title and Abstract to be more results oriented. We have now made a clear distinction between what we have done (i.e. in vitro assays) and how this may apply in cells. We thank the reviewers for prodding us on Bik1. We have now included in the Discussion a potential explanation for the Bik1 phenotype, taking into account results from the Roberts et al. (2014) paper.

We have Bik1 data, which is consistent with our results. However, we have it only for porcine tubulin (not yet for yeast tubulin) and a thorough job requires combination with other proteins such as Bim1 and Kip3, which will greatly increase the scope.

*2) A quite crucial point, that could be addressed experimentally, is whether there is any difference in force-dependence of Kip2/MT interactions at the tips of MTs relative to the lattice. If accumulation at the tip is an important property of Kip2's function, perhaps its ability to withstand loads at the tip is higher than that along the lattice. If the authors elected to carry out this experiment, their model would be supported more strongly.*

The reviewers ask a very interesting question, especially in the context of microtubule dynamics after the growing end has made contact with the bud cortex and starts interacting with dynein. What is the signal to switch to shortening? The reason for including the (lattice) force experiments was to characterize the basic mechanical properties of this motor. Our finding of a low force (like Kip3) is consistent with the idea that Kip2 is a regulator of dynamics that carries small (molecular) cargos and argues against it being a transporter of large organelles (though the reviewers’ later point 5 about many motors is well taken and we have included a statement to this effect).

The requested experiment is exceedingly difficult, as we found out for the non-motor polymerase XMAP215 (see Trushko et al. 2013 PNAS 110, 14670–14675). A thorough job requires testing interactions with other proteins (Bik1, Bim1, cytoplasmic dynein), which again greatly increases the scope. While interesting in its own right, we do not think that the outcome of the requested experiment will significantly affect our principal finding, namely that that Kip2 is a length-dependent polymerase and anti-catastrophe factor.

*Minor points:*

*1) A gel showing the purified Kip2 protein used in the experiments should be included in the primary figure or supplement. I could not find details on the yeast tubulin used – how was this prepared? N.B. the yeast tubulin experiments are a terrific addition to the manuscript.*

We have added a gel of Kip2 and Kip2-GFP in Figure 1—figure supplement 4. We have included a reference to the yeast tubulin preparation; we apologize for the omission.

*2) While it is clear from the kymograph in Figure 2 that Kip2 is a processive motor protein, the authors should present the distribution of run lengths and velocities in a supplementary figure.*

We have added histograms of velocities, run lengths and end-residence times in Figure 2—figure supplement 1.

*3) Figure 1 – previous normal reported growth rate of MTs at 12 μm (at 25 degrees) is 15 nm/s, or 0.9 um/min (Gardner, Zanic, Cell 2011). Why is the growth rate at 0 nM Kip2 half the prior value? Is it a temperature difference between the different experiments or difference in the tubulin preparations?*

We believe that the difference is likely due to the difference in the salt concentrations between the present study and the earlier one. Here we used BRB20 + 100 mM KCl whereas in the Gardner paper BRB80 + 110 KCl was used (the higher salt was necessary to keep Kip3 happy). This is approximately a two-fold difference in ionic strength. There was no temperature difference between the two studies.

*4) Please include the residence time distribution of single Kip2 motors as a supplementary figure. How are the single-molecule plus ends residence times determined if there are multiple Kip2s present? Are shorter MTs used to establish single-molecule conditions at the plus end for step bleaching? How is loss due to dissociation vs. loss due to bleaching discriminated? More technical detail on this should be included in the Methods.*

A histogram of residence times on CPP MTs has been added as Figure 2—figure supplement 1. Residence times were measured at low Kip2-GFP concentrations such that the arrival and departure of individual molecules could be clearly resolved.

In the Methods, we have added a quantification of the Kip2-GFP bleaching rate (measured using AMP-PNP to bind motors to the lattice): it is small and has little effect on the residency time (or on the run length).

*5) Results and Discussion: "The low force argues against a role of Kip2 as an organelle transporter". The authors should tone down this statement. Organelles are often transported by multiple motors, the force measurements are performed on single Kip2 motors (stall force 0.8 pN, reported slip event 3 pN), which could very well be sufficient for organelle transport, especially if teams of motors are operating and if assumed that forces produced by groups of motors are (at least partially) additive. Also, as noted in the major comment above, the effect of Kip2 on microtubule growth appears to be dependent on its cargo Bik1 (not an organelle but still related to transport).*

This is a good point and we restated our argument more carefully, including the possibility that multiple motors could generate larger forces. We also elaborate on the point that Kip2 is a molecule transporter in vitro and in vivo.

6) Figure 4 legend: The authors state: “Thus there was 5 μm of growth at short lengths”. Short lengths are defined < 4 μm. Please revise the text to increase clarity. This section is confusing to read when combined with the figure schematic. Both the schematic and text need revision.

We have revised the text by introducing the term “Catastrophe length” (the average length of a microtubule when it undergoes catastrophe). We believe that the figure is also clearer.

*7) The mechanical properties of Kip2 are likely defined using porcine tubulin. Please mention this in the text.*

We now state in the text that the tweezers experiments used porcine tubulin.

*8) Figure 1—figure supplement 1: Res. frequency is reported in μm-1. Please correct.*

This is corrected.

*9) Table 2: Please add units.*

This is corrected.

*10) The prior work from the Howard group (Varga et al.) on length-dependent control focused on the processive motor Kip3 that induces subunits loss at ends. In a dynamic microtubule assay with kinesin-1 is there any effect on growth rate? Does Kip2 have higher affinity for tubulin dimers – one model would be that the unbound head at the end would help "deliver" a tubulin dimer and thereby accelerate growth (this would not explain the effect on catastrophe, however). Some discussion on what may make Kip2 special based on its catalytic domain/other features would be useful to include.*

It is not known whether kinesin-1 influences dynamic microtubules. Kip2 differs from Kip3 (Kinesin-8) and MCAK (Kinesin-8) (Gardner et al. 2011), which increase catastrophe frequency. Kip2 also differs from Kinesin-1, which has no effect on the rate of depolymerization of GMP-CPP microtubules (Varga et al. 2009).

We do not have definitive results about the relative affinities and the mechanism of growth acceleration and so will not add data on this point. However, we have added a brief discussion on potential molecular mechanisms.